# Pregnant women's intentions to use maternity waiting homes and its associated factors in rural districts of Hadiya Zone, Southern Ethiopia

Habtamu Hasen[1], Getachew Arage[2], Manayeh Mulusew[3], Romedan Delil[3], Ashebir Endale[4], Hassen Mosa[5,6]*, Ritbano Ahmed[5]

1 Department of Emergency Medical Care, Hossana College of Health Sciences, Hossana, Ethiopia, 2 Department of Nutrition and Dietetics, College of Health Science, Debre Tabor University, Debre Tabor, Ethiopia, 3 Department of Clinical Nursing, Hossana College of Health Sciences, Hossana, Ethiopia, 4 Department of Health Extension, Hossana College of Health Sciences, Hossana, Ethiopia, 5 Department of Midwifery, College of Medicine and Health Sciences, Wachemo University, Hossana, Ethiopia, 6 Department of Midwifery, College of Medicine and Health Sciences, Werabe University, Werabe, Ethiopia

* hassenmosa17@gmail.com

**Data Availability Statement:** All relevant data are within the paper and its Supporting Information files.

## Abstract

### Background

As part of a strategy to reduce maternal and perinatal mortality, Ethiopia's government has made a significant effort to expand the number of Maternity Waiting Homes (MWHs). However, worldwide there is a substantial regional variation in pregnant women's intention to use MWHs. Therefore, the aim of this study is to assess pregnant women's intention to use maternity waiting home and its associated factors in the rural district of Hadiya Zone, Southern Ethiopia.

### Methods

This was a cross-sectional study carried out on 385 pregnant women from March 1–28, 2020. A systematic random sampling technique was used to recruit the study participants. SPSS software (version 24.0) was used to enter and analyze the data. Bivariate and multivariate logistic regression analyses were used to determine an association between each independent and dependent variables. Odds ratio with their 95% confidence intervals was computed to ascertain the existence and strength of an association, and statistical significance was affirmed at a $p$-value of < 0.05.

### Results

The prevalence of pregnant women's intension to use MWHs was observed to be 55.6%. A poor wealth status (AOR = 2.52; 95% CI:1.05–6.05), having a previous history of institutional delivery (AOR = 4.78; 95% CI:1.16–9.64), attending four or more antenatal care visits (AOR = 3.34; 95%CI:1.35–8.29), having obstetric complications during previous pregnancy

**Funding:** Hosanna College of Health Science funded the research and it is open for the researchers to publish the manuscript. The funders had no role in study design, data collection and analysis, decision to publish, or preparation of the manuscript.

**Competing interests:** The authors have declared that no competing interests exist.

**Abbreviations:** ANC, Antenatal Care; AOR, Adjusted Odds Ratios; COR, Crude Odds Ratio; CI, Confidence Interval; MWHs, Maternity Waiting Homes; SPSS, Statistical Package for Social Science.

(AOR = 3.76; 95% CI:1.45–9.77), and having favourable attitude towards MWHs (AOR = 13.51; 95% CI: 5.85–9.54) had a significant association with an intention to use MWHs.

## Conclusions

According to the findings of this study, more than half of pregnant women have been intended to use MWHs. Therefore, boosting the uptake of the antenatal care visit, raising awareness about the risk and consequences of obstetric complications, and strengthening behavioral modification strategies is very crucial to increase pregnant women's intention to use MWHs.

## Introduction

The United Nations' Sustainable Development Goal 3 agenda repeats a global commitment to decrease maternal mortality to 70 deaths per 100,000 live births by 2030 [1]. Major progress was made in reducing maternal mortality worldwide over the Millennium Development Goals period, from 1990 to 2015; but, levels remain high, with significant regional differences. In 2015, global maternal mortality was 216 per 100,000 live births, but in sub-Saharan Africa, it was more than double that (546 per 100,000 live births). Ethiopia is one of the ten countries responsible for almost 60% of all maternal fatalities globally [2].

Generally, the objective of diminishing preventable maternal death has yet to be achieved. According to the results of the 2019 Ethiopia Mini Demography and Health Survey (EMDHS), there are 412 maternal deaths per 100,000 live births in Ethiopia. Furthermore, only about half women were delivered by a skilled provider and 34% of women received a postnatal care check-up within the first two days after birth. Regarding antenatal care (ANC), 74% of the women received ANC from skilled provider at least once during their last pregnancy [3].

Maternity Waiting Homes (MWHs) are residential facilities within hospitals or health centers that support pregnant women in their final weeks of pregnancy in order to "bridge the geographical gap" in obstetric care between rural areas with limited access and urban areas with maternity services. It's part of a larger effort backed by the World Health Organization (WHO) to improve access to competent obstetric care and reduce maternal mortality by bringing mothers closer to a health facility. MWHs also reduce home deliveries, which has a negative influence on both women and newborns' health [4–7].

The use of MWHs is associated with a complex variety of risk factors. These factors include socio demographic: having a rich wealthy status [8,9], being poor, longer distance from health facility [10–12], making decision jointly with husbands [10], marital status [13], and being housewives [13,14].

The obstetric factors like: having complications in a previous childbirth [8,10,12,15,16], being primiparous [12], multiparous [17], a positive subjective norm, having a positive attitude towards MWHs and ANC visits are some of the contributing factors of an intention to use MWHs [8,13,18,19]. Additionally, positive behavioural control and perceived benefits of using a MWHs were associated factors of intention to use MWHs [16,18,19].

The Ethiopian Ministry of Health has been deploying MWHs across the country to promote access to high-quality obstetric care since 2014 [20]. However, studies have revealed that there is still a low desire to use MWHs. Moreover, just a few research on women's intentions to use MWHs have been conducted in the region with high inequality of maternal care. For instance, in Jimma and Northwest Ethiopia, the prevalence of pregnant women's intention to

use MWHs varies from 38.7% to 65.3% [8,18]. In Ethiopia generally, and particularly in the study area, the extent and predisposing factors of using MWHs have not yet been fully examined. Therefore, additional study on MWHs would enable program managers and service providers in creating effective interventions and giving pregnant women the best care possible. This study's objective is to examine more about pregnant women's intentions to use MWHs and its contributing factors in rural districts of Hadiya Zone, South Ethiopia.

## Methods and materials

### Study design and period

This was a cross-sectional study carried out from March 1–28, 2020.

### Study setting

The study was undertaken in the rural district of Hadiya Zone, Southern Ethiopia. Hadiya is one of the zones, which found in the Southern part of Ethiopia. It is located 230 kilometers in the Southern part of Addis Ababa, the capital city of Ethiopia, and Northwest direction to Hawassa. According to the 2019/2020 Hadiya Zone Health Department report, the total population was more 1,650,104 of which 830,002 were females. The zone was divided into ten rural Woreda and four administrative towns, comprising 329 kebeles (the lowermost administrative unit of the Ethiopian government system), of which 303 and 26 were respectively rural and urban. It has a total of 311 health posts, 60 rural health centers and four hospitals [21].

The source population embraced all pregnant women who resided in the rural districts of Hadiya zone while, the study population included pregnant women who resided in the selected kebeles during the study period. Participants who were critically ill during the data collection period were excluded from this study.

## Sample size determination

The sample size was determined using single population proportion formula based on the following assumptions; 57.3% the prevalence of intentions to use MWHs among pregnant women which is taken from a previous study [15], 95% confidence interval, margin of error of 5%, and 10% non- response rate. Finally, we have used the finite population correction formula since the source population was 1540 based on the reports from the health extension workers (who is responsible for identifying pregnant women within their catchment area, delivering antenatal care, and connecting them with the formal health system in the event of elevated risk or complications). Consequently, the final sample size was 385. A lottery method was used to pick ten rural kebeles. According to the approximate number of pregnant women in each kebele, the sample size was distributed proportionally. The households within each kebeles were selected by systematic random sampling technique based on the order of the households on the sampling frame obtained from family folders registry with a sampling interval of every four households with the help of health extension workers.

## Measurements

Three items on a five-point Likert scale were used to assess the **intention to use MWHs**. Positive statements were scored in the following order: strongly agree = 5, agree = 4, neutral = 3, strongly disagree = 2, disagree = 1), and negative statements were scored in the opposite order. Participants with a score of 60% or higher were deemed to be **intending to use MWHs** [8]. Four items on a five-point Likert scale were used to assess women's **attitude towards MWHs**.

Participants with a score of 60% or higher were deemed to have a **favourable attitude towards MWHs** [8].

To determine the **subjective norm towards MWHs**. Five items have been used statements that address how important persons (husband, father, mother, and health extension worker) in their lives would perceive their stay at MWH. Five scales ranging from strongly disagree (1) to strongly agree (5). The items were summed to form a direct subjective norm score, and the highest score highest influences of the important reference. Participants with a subjective norm of 60% or higher were assumed to have a **positive subjective norm** [8].

**Health post.**    It is defined as a health facility at the primary level of the care that provides mainly essential promotional and preventive services with limited curative service [22].

**Obstetric complications.**    In this study, obstetric complications include at least one of the following (Postpartum hemorrhage, Eclampsia, Sepsis, and/or Obstructed Labor).

**Gestational age (GA).**    It calculated using the first day of the last menstrual period for women who remembered it and the fundal height to estimate GA for those who didn't.

**Symphysio-fundal height measurement (SFH).**    SFH measurement was undertaken at enrolment using a non-elastic tape measure. Single measurement was taken from the highest point of the uterus (fundus) to the top of the symphysis pubis.

## Last menstrual period (LMP)

As the midwives join the mother into the study, patients were asked about the date of their first day of last menstruation.

**Data collection tools and procedures.**    The questionnaire was initially prepared in English, translated to Hadiyissa and Amharic, then translated back to English to ensure for consistency. Its consistency was checked by translating it back into English by two different experts. Data were collected by Hadiyissa and Amharic language, which is the local language in the study area. Data were collected by pretested and structured questionnaire through face to face interview. Four diploma midwives were recruited for the data collection, and two BSc. midwives participated in supervising the data collectors. Training was given to data collectors and supervisors on the rationale of the study, questionnaire and how to fill the responses. Data was collected by pretested and structured questionnaire through face to face interview. The questionnaire was adapted from interrelated published articles by considering purpose of the study and local circumstances [8–11]. The validity of the questionnaire was approved through the appropriate application of validity criteria (content validity). The reliability test resulted in a Cronbach's alpha value of 0.85.

Information was collected on socio-demographic data; obstetric history, intention to use MWHs and behavioural characteristics. The questionnaire was pretested in non-selected rural kebele of Hadiya zone with 5% of the sample size. The tool's understandability, clarity, and structure were updated after the pretest. The questionnaire was checked daily to ensure its completeness and consistency.

**Data processing and analysis.**    SPSS software (version 24.0) was used to enter and analyze the data. Descriptive statistics, frequency, and proportions were used to summarize the results. We used logistic regression analyses to find out the associated factors of intention to use MWHs. Initially, all explanatory variables were examined using bivariate logistic regression analysis. Then, variables which have $p$-value of $\leq 0.25$ in the bivariate logistic regression analysis were transferred to multivariable logistic regression. The degree of association between the independent and dependent variables was determined using the odds ratio (OR) with 95% confidence intervals. The statistical significance was affirmed at a $p$-value of $< 0.05$. To ensure that the required assumptions for multivariable logistic regressions were met, a Hosmer-

Lemeshow goodness-of-fit test was used. Multicollinerity for interaction between independent variables was measured using the variance inflation factor, which had a value of less than five.

The household wealth variable was created using principal components analysis of items listed in below. Items were selected to minimize clustering and truncation which compromise reliability. An asset-based wealth index created using information on asset ownership (radio, television, mobile phone, motorbike, car/truck), number of animals owned (cows, sheep, poultry), electricity supply to home, health insurance, drinking water source, type of toilet and type materials used for construction of floors in the home. It was divided into three groups: low (poor), medium, and high (rich) [23].

## Ethics approval and consent to participate

The Institutional Review Board of the Hossana College of Health Sciences gave ethical approval. The Hadiya Zone Health Office was then asked for approval before the data collection process was begin. The participants were fully informed about the purpose, methods, potential dangers, and advantages of the study. Selected individuals were asked to give written informed consent prior to the interview to ensure their willingness to participate in the study. Respondents under the age of 18 requested parental or guardian consent. In order to ensure confidentially, the respondent's name was omitted from the written questionnaire. Additionally, participants were informed that their care would not change if they declined to agree or withdrew from the study.

## Results

### Sociodemographic characteristics

A total of 385 study participants were completed the interview, yielding a response rate of 100%. The mean age of study participants was 28 (±5.02) years. The majority of the participants were in union 377(97.9%), 345 (89.6%) were Hadiya in ethnicity, 315(81.7%) were protestants in terms of their religion, and 284(73.8%) were housewives. In terms of educational attainment, 200(51.9%) of participants had completed primary school, and 129(33.5%) had a low socioeconomic status (**Table 1**).

### Obstetrics characteristics

Among the participants, 240(62.4%) of them were multiparous, 80(20.8%) had ever had an abortion, and 274(71.2%) had not encountered an obstetric complications. Besides, 268 (69.8%) had four or more ANC visits, 329 (87.7%) gave birth to their last child at the health facility and 256(66.5%) heard about MWHs (**Table 2**).

### Intentions to use MWHs and behavioural characteristics

In this study, 214 (55.6%) of the study participants had an intention to use MWHs. Moreover, 184 (85.6%) had a favourable attitude, while163 (77.6%) had a positive subjective norm.

### Factors associated with intentions to use MWHs

After controlling for possible confounders using the multivariate logistic regression analysis model, attending four or more ANC visits, poor wealth status, having previous institutional delivery, having complications in the previous pregnancy, and having a favourable attitude towards MWHs were significantly associated with an intention to use MWHs.

The odds of intention to use MWHs were 2.52 times higher among poor pregnant women when compared with their counterparts (AOR = 2.52; 95% CI: 1.05–6.05).

**Table 1. Socio-demographic characteristic of study subjects in rural districts of Hadiya Zone, Southern Ethiopia, March 2020.**

| Variables | Numbers (n = 385) | Percent |
|---|---|---|
| **Age in years** | | |
| ≤ 24 | 74 | 19.2 |
| 25–29 | 159 | 41.3 |
| ≥30 | 152 | 39.5 |
| **Marital status** | | |
| In union | 377 | 97.9 |
| Not in union | 8 | 2.1 |
| **Religion** | | |
| Protestant | 315 | 81.7 |
| Orthodox | 33 | 8.6 |
| Muslim | 26 | 6.8 |
| Catholic | 11 | 2.9 |
| **Ethnicity** | | |
| Hadiya | 345 | 89.6 |
| Kambata | 19 | 4.9 |
| Silte | 12 | 3.2 |
| Amhara | 9 | 2.3 |
| **Educational level of the mother** | | |
| No formal education | 85 | 22.0 |
| Primary education | 300 | 78.0 |
| **Occupational status of the mother** | | |
| Housewives | 284 | 73.8 |
| Farmer | 43 | 11.2 |
| Merchant | 24 | 6.2 |
| Government employee | 34 | 8.8 |
| **Educational level of husband** | | |
| No formal education | 41 | 10.8 |
| Primary education | 189 | 49.0 |
| Secondary education | 105 | 22.3 |
| College and above | 50 | 12.9 |
| **Occupation of husband** | | |
| Farmer | 234 | 60.8 |
| Merchant | 90 | 23.4 |
| Government employee | 47 | 12.2 |
| Daily laborer | 14 | 3.6 |
| **Time to reach the nearest health facility by walking on foot in minute** | | |
| <30 | 215 | 55.8 |
| ≥30 | 170 | 44.2 |
| **Number of household members** | | |
| <5 | 160 | 41.6 |
| ≥ 5 | 225 | 58.4 |
| **Wealth status** | | |
| Rich | 134 | 34.8 |
| Medium | 122 | 31.7 |
| Poor | 129 | 33.5 |

**Table 2. Obstetrics characteristics of study subjects in rural districts of Hadiya Zone, Southern Ethiopia, March 2020.**

| Variables | Numbers(n = 385) | Percent |
|---|---|---|
| **Ever had abortion** | | |
| yes | 80 | 20.8 |
| No | 305 | 79.2 |
| **Ever had stillbirth** | | |
| Yes | 20 | 5.2 |
| No | 365 | 94.8 |
| **Previous cesarean delivery** | | |
| Yes | 47 | 12.2 |
| No | 338 | 87.8 |
| **Number of ANC visit for the current pregnancy** | | |
| <4 | 117 | 30.4 |
| ≥ 4 | 268 | 69.6 |
| **Gestational age in weeks** | | |
| First trimester (<14) | 91 | 23.6 |
| Second trimester (14–28) | 132 | 34.3 |
| Third trimester(28-delivery) | 162 | 42.1 |
| **Age at first pregnancy** | | |
| ≤20 | 272 | 70.6 |
| ≥21 | 113 | 29.4 |
| **Parity** | | |
| Nullipara | 10 | 2.6 |
| Primipara | 135 | 35 |
| Multipara | 240 | 62.4 |
| **Heard about MWHs** | | |
| Yes | 256 | 66.5 |
| No | 129 | 33.5 |
| **Having a previous obstetric complications during pregnancy** | | |
| Yes | 111 | 28.8 |
| No | 274 | 71.2 |
| **Prior place of delivery** | | |
| Home | 46 | 12.3 |
| Health facility | 329 | 87.7 |
| **Overall intentions to use MWHs** | | |
| Yes | 214 | 55.6 |
| No | 171 | 44.4 |

Similarly, the odds of intention to use MWHs were 3.34 times higher among pregnant women who had ≥ 4 ANC visits as compared to those women who had < 4 ANC visits (AOR = 3.34; 95% CI:1.35–8.29).

Likewise, the odds of intention to use MWHs were 4.78 times higher among pregnant women who delivered in health institution when compared to pregnant women who delivered at home(AOR = 4.78; 95% CI: 1.16–9.64).

Additionally, those women who have encountered a complication during their previous pregnancy were 3.76 times more likely for having an intention to use MWHs during their next pregnancy as compared to their counterparts (AOR = 3.76; 95% CI: 1.45–9.77).

**Table 3. Factors associated with pregnant women's intention to use MWHs in rural districts of Hadiya Zone, Southern Ethiopia, March 2020 (n = 385).**

| Variables | Intention to use MWHs | | COR (95% CI) | AOR (95% CI) |
|---|---|---|---|---|
| | Yes | No | | |
| **Age in years** | | | | |
| ≤24 | 29 (39.2) | 45 (60.8) | Reference | Reference |
| 25–29 | 79 (49.7) | 80 (50.3) | 1.53(0.87–2.68) | 1.33 (0.65–3.94) |
| ≥ 30 | 106 (69.7) | 46(30.3) | 3.57(2.34–6.39)* | 3.52 (0.83–9.71) |
| **Educational status of the mother** | | | | |
| Formal education | 50(58.8) | 35(41.2) | Reference | Reference |
| No Formal education | 164(54.6) | 136(45.4) | 0.8.(0.9–7.4) | 0.75(0.6–7.2) |
| **Distance to health facility** | | | | |
| <30 | 70(32.6) | 145(67.4.) | Reference | Reference |
| ≥30 | 90(52.9) | 80(47.1) | 2.3(0.83–8.65) | 2.2(0.76–6.45) |
| **Wealth status** | | | | |
| Rich | 94(70.1) | 40(29.9) | Reference | Reference |
| Medium | 71(58.2) | 51(41.8) | 2.27(1.37–3.76) | 1.56 (0.65–3.69) |
| Poor | 49(38.0) | 80(62.0) | 3.84(2.29–6.41)* | 2.52 (1.05–6.05)** |
| **Number of ANC visit** | | | | |
| < 4 | 42(35.9) | 75 (64.1) | Reference | Reference |
| ≥ 4 | 96 (35.8) | 172(64.2) | 3.19(2.03–5.03)* | 3.34 (1.35–8.29) ** |
| **Previous place of delivery** | | | | |
| Home | 39(84.8) | 7(15.2) | Reference | Reference |
| Health facility | 125(38.0) | 204(62.0) | 9.09(3.94–13.95)* | 4.78 (1.16–9.64) ** |
| **Previous use of MWHs** | | | | |
| No | 157(85.8) | 26(14.2) | Reference | Reference |
| Yes | 57(28.2) | 145(71.8) | 15.36(9.17–13.73)* | 2.26 (0.74–4.51) |
| **Having complications during previous pregnancy** | | | | |
| No | 200(73.0) | 74 (27.0) | Reference | Reference |
| Yes | 14(12.6) | 97 (87.4) | 18.73(10.06–19.83)* | 3.76 (1.45–9.77)** |
| **Attitude towards MWHs** | | | | |
| Unfavorable | 30(16.6) | 151(83.4) | Reference | Reference |
| Favorable | 184(90.2) | 20 (9.8) | 14.31(11.27–16.82) | 13.51(5.85–9.54)** |
| **Subjective norm** | | | | |
| Negative | 51 (29.1) | 124(70.9) | Reference | Reference |
| Positive | 163(77.6) | 47(22.4) | 8.43 (5.32–13.35)* | 2.81(0.68–6.16) |

* = p ≤ 0.25

** = p < 0.05.

Moreover, the odds of intention to use MWHs were 13.51times higher among pregnant women who had a favourable attitude towards MWHs as compared to their counterparts (AOR = 13.51; 95% CI: 5.85–9.54) (**Table 3**).

## Discussion

In this study, the intention of pregnant women to use MWHs was found to be 55.6%. This is higher than the 38.7, 31.5, and 45% found in studies conducted in Jimma, Bench Maji Zone, and Kenya, respectively [18–20]. This discrepancy may be due to differences in the study area, sociocultural backgrounds, or different research strategies. However, this prevalence was lower than the 65.3 and 74.3% recorded in Northwest Ethiopia and Debra Markos town,

respectively [8,24]. The discrepancy may be explained by variations in the sample size, study period, culture, and demography. The propensity to use MWHs can also vary between and within geographical regions.

According to the findings of the current study, poor pregnant women have 2.52 times higher probabilities of intending to use MWHs than their counterparts. This finding is in line with a study carried out in Tanzania [11]. The MWHs may provide poorer women with access to higher level obstetric care without having to pay for private transportation during labor and delivery, which could be the cause. However, this result is in contrast with the studies conducted in Northwest Ethiopia, Jimma zone, and Timor Leste, respectively [8,9,25]. The richest women may have the financial means to be transported to the hospital immediately when labor begins.

In this study, pregnant women who had four or more ANC visits had 3.34 times greater chances of intending to utilize MWHs than those who had less than four ANC visits. This result is consistent with previous research conducted in the northwest Ethiopia, Digelu and Tijo district, and Bench Maji Zone [8,17,19]. This may be because pregnant women who had more ANC visits were more likely to be aware of pregnancy and childbirth danger signs. As a result, they can conveniently opt to remain at the health facility prior to giving birth.

Distance remains an impending obstacle to healthcare access. However, utilization of MWHs act as a bridge to skilled care by providing temporary shelter near a facility staffed by professionals. In this study, living at a distance that taken a time 30 min and more from health facility was not significantly associated with utilization of MWHs. However, this finding is inconsistent with studies conducted in Zambia and Malawi [12,16]. The probable explanation comprises, difference in socioeconomic status, infrastructure, media advertisement about use of MWHs, and implementation of MWHs.

In addition, the current study found that women who had a problem during their prior pregnancy were 3.76 times more likely than their counterparts to intend to utilize MWHs during their next pregnancy. This finding is consistent with research conducted in Arba Minch, Malawi, and rural Zambia [10,12,16]. The reason could be women who have experienced a birth complication may be afraid that they will experience similar incidents during their next childbirth, so they choose to live in MWHs to be close to health facilities. In contrast, a study conducted in Ethiopia's Gurage zone [26] found that having an obstetric complication has no significant association with the intention to use MWHs. This may be due to variations in the study area, since the present study was conducted entirely in a rural district, where the study population might not have had any options for getting to health facilities.

The current study also indicated that pregnant women who delivered in a health facility had 4.78 times higher odds of intending to use MWHs than pregnant women who delivered at home. This finding is supported by research conducted in Arba Minch, Malawi, and rural Zambia [10,12,16]. The possible explanation might be women who have previously had an institutional delivery will be more likely to receive proper counselling and advice from their health care providers during subsequent deliveries if they have previously had an institutional delivery at MWHs. As a result, their previous experiences may have influenced their decision to use MWHs.

According to this study, the odds of intending to use MWHs were 13.51 times greater among pregnant women who had a favorable attitude toward MWHs than their counterparts. This finding is consistent with the result of studies conducted in Northwest Ethiopia, Southwest Ethiopia, Jimma, and Bench Maji Zone [8,18,19]. It's possible that the observed association is due to initiatives that foster a positive attitude toward MWHs and therefore have the potential to increase service use. The present study's findings, however, contradicted those of a

study conducted in the Mettu district [27]. The dissimilarity could be attributed to differences in sample size, study period and study settings.

In this study, maternal educational level was not significantly associated with MWHs use. The discrepancy could be explained by the Ethiopian government's strong commitment to expanding MWHs program distribution through the mobilization of health extension workers and the women's health developmental army in lobbying for maternal health services at grass root level.

One of the study's strengths is that the study participants were recruited using a probability sampling method to ensure the study's representativeness, and those different techniques were used to maintain data quality. But, as a drawback, this research has the limitations of a cross-sectional study. A qualitative method was not used to aid this study, either. Furthermore, the primary outcome was focused on women's self-reported MWHs use, which may be prone to recall and social desirability bias. However, because remaining at MWHs prior to delivery is supposed to be a noticeable event, this risk is thought to be small.

## Conclusions

In conclusion, more than half of pregnant women had an intention to use MWHs. The intention to use MWHs was significantly associated with poor wealth status, attending four or more ANC visits, having a previous history of institutional delivery, a favourable attitude towards MWHs and having complications during previous pregnancy. Therefore, boosting the uptake of the ANC visit, raising awareness about the risk and severity of obstetric complications, and strengthening behavioral modification strategies is very crucial to increase women's intention to use MWHs. Future research should be conducted using a mixed method study design to better understand the specific reasons for not intending to use MWHs.

## Supporting information

**S1 File. Informed consent and questionnaire.**
(DOCX)

**S2 File. SPSS.**
(SAV)

## Acknowledgments

We would like to thank Hossana College of Health Science for providing ethical approval. Additionally, the district of Hadiya Zone Health Office, study participants, data collectors, and supervisors all deserve our sincere gratitude.

## Author Contributions

**Conceptualization:** Habtamu Hasen.

**Data curation:** Habtamu Hasen, Getachew Arage, Romedan Delil, Ashebir Endale, Hassen Mosa.

**Formal analysis:** Habtamu Hasen.

**Funding acquisition:** Habtamu Hasen.

**Investigation:** Habtamu Hasen.

**Methodology:** Habtamu Hasen, Manayeh Mulusew, Ashebir Endale, Hassen Mosa, Ritbano Ahmed.

**Project administration:** Habtamu Hasen, Romedan Delil, Ashebir Endale, Hassen Mosa.

**Resources:** Habtamu Hasen.

**Software:** Habtamu Hasen, Ritbano Ahmed.

**Supervision:** Getachew Arage, Manayeh Mulusew, Romedan Delil, Ashebir Endale, Hassen Mosa, Ritbano Ahmed.

**Validation:** Getachew Arage, Manayeh Mulusew, Romedan Delil, Ashebir Endale, Hassen Mosa, Ritbano Ahmed.

**Visualization:** Getachew Arage, Manayeh Mulusew, Romedan Delil, Ashebir Endale, Hassen Mosa, Ritbano Ahmed.

**Writing – original draft:** Habtamu Hasen, Hassen Mosa.

**Writing – review & editing:** Habtamu Hasen, Getachew Arage, Manayeh Mulusew, Romedan Delil, Hassen Mosa, Ritbano Ahmed.

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
