## [Decision Letter · Decision Letter 0]

5 Aug 2021

PONE-D-21-13771

Facilitators and barriers of pregnant women’s intention to use maternal waiting home in Southern Nations, Nationalities and Peoples Region, Ethiopia

PLOS ONE

Dear Dr. Mosa,

Thank you for submitting your manuscript to PLOS ONE. After careful consideration, we feel that it has merit but does not fully meet PLOS ONE’s publication criteria as it currently stands. Therefore, we invite you to submit a revised version of the manuscript that addresses the points raised during the review process.

We look forward to receiving your revised manuscript.

Kind regards,

Michelle L. Munro-Kramer, PhD, CNM, FNP-BC

Academic Editor

PLOS ONE

Journal Requirements:

3. Please amend the manuscript submission data (via Edit Submission) to include author Manayeh Mulusew.

Additional Editor Comments (if provided):

Thank you for submitting your manuscript on this important topic. Both reviewers noted that this manuscript could be a substantial contribution to the literature, while also identifying numerous areas where clarification and copy editing are necessary. I recommend that the authors carefully attend to each reviewer comment. In addition to the reviewer comments, I would also suggest:

1) Page 12, lines 100-102, providing some context/definitions on what the different administrative regions mean (e..g, kebeles, health posts)

2) Page 13, lines 142-143 & Page 14, lines 152-156, more details are needed on the training for data collectors. How were they trained and what does it mean that they had close supervision? Did the data collectors complete the survey or did participants complete the survey or did it depend on literacy?

3) I also concur with Reviewer #1, that submitting the questionnaire is very valuable, but not all readers will review it. It is therefore important to provide examples of the survey items within the text.

We look forward to receiving your revised manuscript.

Reviewers' comments:

Reviewer's Responses to Questions

**Comments to the Author**

1. Is the manuscript technically sound, and do the data support the conclusions?

Reviewer #1: Yes

Reviewer #2: Yes

2. Has the statistical analysis been performed appropriately and rigorously? 

Reviewer #1: Yes

Reviewer #2: Yes

3. Have the authors made all data underlying the findings in their manuscript fully available?

Reviewer #1: Yes

Reviewer #2: Yes

4. Is the manuscript presented in an intelligible fashion and written in standard English?

Reviewer #1: Yes

Reviewer #2: No

5. Review Comments to the Author

Reviewer #1: This article adds to the growing literature being published on maternity waiting homes and adds specific insight relevant to the Ethiopian context. Additionally, it adds to the specifically scarce literature on intention to use MWHs. The authors provide sufficient justification for the research as there is currently no information on intention to use MWHs in the specific region of Ethiopia in which the research was conducted. The article generally appears scientifically sound and the conclusions are supported by the data. Generally, the writing is clear and the processes are easy to follow. The authors cite relevant literature about MWHs. I have a number of major and minor ones to be addressed before publication of the article should be considered.

Major revisions

1) Title: The title of the article is inaccurate. This article is not about barriers and facilitators to pregnant women’s intention to use MWHs. It is about the factors associated with that intention – Women’s individual-level characteristics and their obstetric history. Other articles have been published about the barriers and facilitators to use MWHs and usually have asked women qualitatively what their experience has been.

2) Premise of article: All of the questions in the survey ask specifically about intention to use MWHs for 15 days but this is not brought up in the methods, discussion, or limitations. This aspect needs to be addressed. It could be considered a major limitation of the study as women who may have intended to use the MWH for only 10 days or even for 18 days, may have answered all of the intention to use questions in the negative because of how specific they were. While 15 days may be recommended by the Ethiopian Ministry of Health (assumption), it does not mean all women will follow this directive to the letter. I suggest including the 15 day aspect in the article title, explaining it in the methods, including it in table titles and subheadings, and discussing the implications of the 15 days in the discussion, including as a limitation. Include an explanation on why the 15 days was specifically chosen, how it relates to government policy or other MWH use guidelines if at all, and how it relates to the average length of stay of MWH users in Ethiopia and elsewhere.

3) Introduction: Paragraph 2 of the introduction (or a small additional paragraph) should include findings to date on the impact of MWHs on maternal and neonatal health outcomes. While an important study in Ethiopia was cited, there are other articles which address the impact of MWH in sub-Saharan Africa on maternal mortality, facility delivery rates, stillbirths, and caesarian section rates. Lines 86-87 should be moved to this paragraph and include those citations.

4) Introduction: The last sentence of the introduction is passive. I suggest making it active and engaging. For example: “…sustainable development by providing relevant MWH information and advice to policy makers and implementers in Ethiopia.” See my point below on weaving this into the discussion.

5) Methods: When discussing sampling, clarify and simplify the description. The sentences jump back and forth between source and sample populations. Describe the linear process followed from identifying the source population to selecting the sample

6) Methods: How were the education categories in the questionnaire chosen? These do not appear to be standard, such as what is used in the Demographic and Health Surveys. The authors should include a citation if they have one.

7) Methods: Give examples of questions included in the survey for intention to use MWHs, attitudes toward MWHs, and subjective norms toward MWHs. Not all readers will examine the supplemental questionnaire, so having an example of each in the text would be helpful. However, I appreciate the authors including the questionnaire, it was helpful to have.

8) Methods: Was gestation age considered in the exclusion criteria or in the analysis? It was asked in the survey but not included in the tables. What if the woman had not considered use of the MWH because they were only a few weeks pregnant? Gestational age of the respondents should be included in Table 1 and in the linear regression model if significantly associated with MWH intention.

9) Methods: In the questionnaire, all behavioral belief and outcome evaluation questions were written in the positive and many sound leading. There are no negative views of the MWHs for participants to agree or disagree with. Include an explanation in the methods for how these questions were developed and why the authors chose to write them all in the positive. The authors should include discussion around the possibility of social desirability bias and acquiescence bias in the limitations section. Were the midwives who collected the data from within the health facilities accessible to the population? This could also increase social desirability bias if the participants knew them.

10) Methods: The article would benefit from having a study setting to provide more information on the health system in this region of Ethiopia and the state of MWHs. Since the government is encouraging MWH use, has it also built new MWH structures or are the structures community constructed? Are MWHs only located at primary health centers, only at hospitals, or both?

11) Discussion: In the first sentence of the discussion (lines 262-263), the authors write that the purpose of the research was to find out what factors influenced pregnant women’s decisions to use MWHs. This is not supported by the results. The authors only know what demographic characteristics about the women and their previous obstetric history are associated with intention to use MWHs. They do not know what goes into women’s decision-making regarding barriers they face to access, desire for a safe delivery, have permission or support from their partner, etc.

12) Discussion: The discussion is missing policy or implementer implications. There is a sentence about this in the conclusion that should be included in the discussion and expanded upon. What this research means in terms of suggestions for what Ethiopia’s national strategy regarding MWHs is a critical missing piece.

13) The article requires English language editing and revisions specifically for sentence structure and grammar, particularly verb tense. The writing could generally be tightened up and shortened. The sampling and “factors associated with intentions to use MWHs” are two particular examples of sections that could be tightened up.

14) It would be helpful to provide the location of a cvs file for the data in addition to or instead of the SPSS file as not everyone uses or has access to SPSS. A csv file would allow even users with free software (such as R) to analyze the data. This does not need to be included as a supplementary file, though it can be. It could be housed in a data repository.

Minor revisions

1) General: Consistently use maternity waiting homes throughout the article and the title. The authors switch between maternal and maternity throughout the text when discussing the waiting homes.

2) Abstract, Conclusions: “Almost half” should say “More than half”

3) Key words: Use key words not already included in the title of the paper to reach more readers. I suggest: obstetric care, maternal health, maternal waiting home, sub-Saharan Africa

4) Introduction: What does the sentence beginning paragraph 3 mean? While the list of factors associated with MWH use is technically accurate, it is not engaging for the reader. Consider breaking it down into smaller sentences with logical groupings. For example, the authors could group by individual, interpersonal, and health systems factors.

5) Methods: In the text and tables, clarify which pregnancy the questionnaire asked about. For example, for “Number of ANC visits”, it is unclear whether this is about the current pregnancy or the last one.

6) Methods: How were the reference categories chosen in Table 3. Clarify in the methods.

7) Methods: Lines 124 through 126 on principal component analysis should be moved to the analysis section and include a citation.

8) Methods: I am unclear what the sentence on lines 126 to 127 means. Please clarify.

9) Methods: Lines 151 to 152: “The questionnaire’s validity and reliability was determined in a scientific manner.” The meaning of this sentence is unclear.

10) Table 1: How was time to reach health facility determined and by what transport method? Were women asked time without reference to transport method? Clarify in the methods and in the indicator title in Table 1.

11) Table 1: What does “Number of households” mean? Does this mean number of household members? Clarify.

12) Results: Include the table reference in parentheses, in the first sentence in which data from the table is described. For example, the reference to Table 3 should be in the first sentence of the sub-section, not the last.

13) Discussion: include citations for known barriers to accessing health facilities (lines 276-278)

Reviewer #2: Because PLOS ONE does not copyedit accepted manuscripts, the authors should employ an editor to assist with ambiguous and grammatical errors that appear throughout the text. There are multiple grammar and sentence structure corrections that are required prior to publication.

6. PLOS authors have the option to publish the peer review history of their article (what does this mean?). If published, this will include your full peer review and any attached files.

Reviewer #1: No

Reviewer #2: No

---

## [Author Response · Author response to Decision Letter 0]

16 Sep 2021

Thank you very much for your detail questions, comments and suggestions

Response to the editor

Response to editor: We thankful for the given comment. Now we have tried to follow the PLoS one submission guideline format. 

2. In your Data Availability statement, you have not specified where the minimal data set underlying the results described in your manuscript can be found. 

Response to editor: Thank you for the comment. Now we have clearly described about the Data Availability statement. The editor may this part.

3. Please amend the manuscript submission data (via Edit Submission) to include author Manayeh Mulusew. 

Response to editor: Thank you for the comment. Now we have corrected it according to the given comment.

Additional Editor Comments 

1) Page 12, lines 100-102, providing some context/definitions on what the different administrative regions mean (e.g, kebeles, health posts) 

Response to editor: Thank you for the comment. We share the raised concern. For more clarification we have added the definition of kebeles and health post. The reviewer may see the measurement part.

2) Page 13, lines 142-143 & Page 14, lines 152-156, more details are needed on the training for data collectors. How were they trained and what does it mean that they had close supervision? Did the data collectors complete the survey or did participants complete the survey or did it depend on literacy?

Response to editor: Thank you for the comment. Now we have made a great modification by adding statements on the result and tool part. The reviewer may see the result and the data collection tool part. 

3) I also concur with Reviewer #1, that submitting the questionnaire is very valuable, but not all readers will review it. It is therefore important to provide examples of the survey items within the text.

Response to editor: Thank you for the comment.

1) Title: The title of the article is inaccurate. This article is not about barriers and facilitators to pregnant women’s intention to use MWHs. It is about the factors associated with that intention – Women’s individual-level characteristics and their obstetric history. Other articles have been published about the barriers and facilitators to use MWHs and usually have asked women qualitatively what their experience has been.

Response to reviewer; We are grateful for the given comment and we have made the essential modification on the title. The reviewer may see the title part.

2) Premise of article: All of the questions in the survey ask specifically about intention to use MWHs for 15 days but this is not brought up in the methods, discussion, or limitations. This aspect needs to be addressed. It could be considered a major limitation of the study as women who may have intended to use the MWH for only 10 days or even for 18 days, may have answered all of the intention to use questions in the negative because of how specific they were. While 15 days may be recommended by the Ethiopian Ministry of Health (assumption), it does not mean all women will follow this directive to the letter. I suggest including the 15 day aspect in the article title, explaining it in the methods, including it in table titles and subheadings, and discussing the implications of the 15 days in the discussion, including as a limitation. Include an explanation on why the 15 days was specifically chosen, how it relates to government policy or other MWH use guidelines if at all, and how it relates to the average length of stay of MWH users in Ethiopia and elsewhere. 

Response to reviewers; Thank you for your deep insight. We understand the raised concern. But, to answer the question we have used the WHO statement about Maternity homes were set up in the vicinity of hospitals so that women from areas that were remote and/or difficult to access could be accommodated during the last two weeks or average of 15 days of pregnancy. 

3) Introduction: Paragraph 2 of the introduction (or a small additional paragraph) should include findings to date on the impact of MWHs on maternal and neonatal health outcomes. While an important study in Ethiopia was cited, there are other articles which address the impact of MWH in sub-Saharan Africa on maternal mortality, facility delivery rates, stillbirths, and caesarean section rates. Lines 86-87 should be moved to this paragraph and include those citations. 

Response to reviewer 1: Thank you for the comment. Correction is made according to the raised concern. The reviewer may see the introduction part.

4) Introduction: The last sentence of the introduction is passive. I suggest making it active and engaging. For example: “…sustainable development by providing relevant MWH information and advice to policy makers and implementers in Ethiopia.” See my point below on weaving this into the discussion.

Response to reviewer 1: Thank you for the comment. Correction is made according to the provided comment. Now we make it active. The reviewer may see the introduction part.

5) Methods: When discussing sampling, clarify and simplify the description. The sentences jump back and forth between source and sample populations. Describe the linear process followed from identifying the source population to selecting the sample

Response to reviewer 1: Thank you for the comment. Now we have made a considerable amendment. The reviewer may see the method part.

6) Methods: How were the education categories in the questionnaire chosen? These do not appear to be standard, such as what is used in the Demographic and Health Surveys. The authors should include a citation if they have one.

Response to reviewer: Thank you for the comment. Now we have corrected it according to standard classification of Demographic and Health Surveys. The reviewer may see table 1.

7) Methods: Give examples of questions included in the survey for intention to use MWHs, attitudes toward MWHs, and subjective norms toward MWHs. Not all readers will examine the supplemental questionnaire, so having an example of each in the text would be helpful. However, I appreciate the authors including the questionnaire, it was helpful to have.

Response to reviewer: Thank you for the comment. Now we have included the necessary elements.

8) Methods: Was gestation age considered in the exclusion criteria or in the analysis? It was asked in the survey but not included in the tables. What if the woman had not considered use of the MWH because they were only a few weeks pregnant? Gestational age of the respondents should be included in Table 1 and in the linear regression model if significantly associated with MWH intention.

Response to reviewer: Thank you for the comment. We understand the raised concern. We have made important corrections.

9) Methods: In the questionnaire, all behavioural belief and outcome evaluation questions were written in the positive and many sound leading. There are no negative views of the MWHs for participants to agree or disagree with. Include an explanation in the methods for how these questions were developed and why the authors chose to write them all in the positive. The authors should include discussion around the possibility of social desirability bias and acquiescence bias in the limitations section. Were the midwives who collected the data from within the health facilities accessible to the population? This could also increase social desirability bias if the participants knew them.

Response to reviewer: Thank you for the comment. Now the data collection tool is substantially modified to increase its clarity. The reviewer may see the method part. 

10) Methods: The article would benefit from having a study setting to provide more information on the health system in this region of Ethiopia and the state of MWHs. Since the government is encouraging MWH use, has it also built new MWH structures or are the structures community constructed? Are MWHs only located at primary health centers, only at hospitals, or both?

Response to reviewer: Thank you for the comment. We understand the raised concern. In the tool part we want to say health facility not health center. However, MWHs are located both in health centers and hospitals. Generally, MWHs have been set up near hospitals with no facilities for operative deliveries, or near district or teaching hospitals with operative facilities. The set up chosen will depend on the

11) Discussion: In the first sentence of the discussion (lines 262-263), the authors write that the purpose of the research was to find out what factors influenced pregnant women’s decisions to use MWHs. This is not supported by the results. The authors only know what demographic characteristics about the women and their previous obstetric history are associated with intention to use MWHs. They do not know what goes into women’s decision-making regarding barriers they face to access, desire for a safe delivery, have permission or support from their partner, etc. 

Response to reviewer: Thank you for the comment. We are indebted for the comment and we have removed unnecessary statements. The reviewer may see the introduction part of the discussion.

12) Discussion: The discussion is missing policy or implementer implications. There is a sentence about this in the conclusion that should be included in the discussion and expanded upon. What this research means in terms of suggestions for what Ethiopia’s national strategy regarding MWHs is a critical missing piece. Thank you for the comment. Based on the given comment the essential correction is made. The reviewer may see the discussion part.

Response to reviewer: Thank you for the comment.

13) The article requires English language editing and revisions specifically for sentence structure and grammar, particularly verb tense. The writing could generally be tightened up and shortened. The sampling and “factors associated with intentions to use MWHs” are two particular examples of sections that could be tightened up. 

Response to reviewer: We are grateful the suggestion. Now we have used professional editing service like true editors and proofreading to correct spelling, punctuation and grammatical errors. Moreover, now we tried make it short and precise.

14) It would be helpful to provide the location of a cvs file for the data in addition to or instead of the SPSS file as not everyone uses or has access to SPSS. A csv file would allow even users with free software (such as R) to analyze the data. This does not need to be included as a supplementary file, though it can be. It could be housed in a data repository.

Response to reviewer: We are grateful for the suggestion. We will try to use data repository.

Minor revisions

1) General: Consistently use maternity waiting homes throughout the article and the title. The authors switch between maternal and maternity throughout the text when discussing the waiting homes.

Response to reviewer: Thank you for the comment. Now we have made the necessary amendment.

2) Abstract, Conclusions: “Almost half” should say “More than half”

Response to reviewer: Thank you for the comment. Now it is corrected as per the provided comment. The reviewer may see the abstract part.

3) Key words: Use key words not already included in the title of the paper to reach more readers. I suggest: obstetric care, maternal health, maternal waiting home, sub-Saharan Africa

Response to reviewer: Thank you for the comment. We accept the suggestion and we have an amendment. The reviewer may see the abstract part.

4) Introduction: What does the sentence beginning paragraph 3 mean? While the list of factors associated with MWH use is technically accurate, it is not engaging for the reader. Consider breaking it down into smaller sentences with logical groupings. For example, the authors could group by individual, interpersonal, and health systems factors.

Response to reviewer: Thank you for the comment. We understand the comment and we have modified it accordingly.

5) Methods: In the text and tables, clarify which pregnancy the questionnaire asked about. For example, for “Number of ANC visits”, it is unclear whether this is about the current pregnancy or the last one.

Response to reviewer: Thank you for the comment. Now we have corrected it. The reviewer may see table 2.

6) Methods: How were the reference categories chosen in Table 3. 

Response to reviewer: Thank you for the comment. The reference was selected based on the nature of variables.

7) Methods: Lines 124 through 126 on principal component analysis should be moved to the analysis section and include a citation.

Response to reviewer: Thank you for the comment. We have made the necessary changes.

8) Methods: I am unclear what the sentence on lines 126 to 127 means. Please clarify.

Response to reviewer: Thank you for the comment. We have made a great modification in the method part as a whole. The reviewer may see the method part.

9) Methods: Lines 151 to 152: “The questionnaire’s validity and reliability was determined in a scientific manner.” The meaning of this sentence is unclear.

Response to reviewer: Thank you for the comment. No we have made substantial change regarding to the questionnaire. Additionally, we have removed vague statements. The reviewer may see the method part

10) Table 1: How was time to reach health facility determined and by what transport method? Were women asked times without reference to transport method? Clarify in the methods and in the indicator title in Table 1.

Response to reviewer: We are thankful for the comment. Now we have made the essential change. The reviewer may see Table 1.

11) Table 1: What does “Number of households” mean? Does this mean number of household members? Clarify.

Response to reviewer: Thank you for the comment. Now we have made the necessary change the reviewer may see table 1.

12) Results: Include the table reference in parentheses, in the first sentence in which data from the table is described. For example, the reference to Table 3 should be in the first sentence of the sub-section, not the last. 

Response to reviewer: Thank you for the suggestion. Now we have made the required change the reviewer may see table 3.

13) Discussion: include citations for known barriers to accessing health facilities (lines 276-278)

 Response to reviewer: we are thankful you for the provided comment. Now we have made the required change the reviewer may see the discussion part.

Reviewer #2: Because PLOS ONE does not copyedit accepted manuscripts; the authors should employ an editor to assist with ambiguous and grammatical errors that appear throughout the text. There are multiple grammar and sentence structure corrections that are required prior to publication.

Response to reviewer 2: Thank you for the comment. Now we have used professional editing services like paper true and proofreading to correct spelling, punctuation and grammatical errors.

---

## [Decision Letter · Decision Letter 1]

30 Mar 2022

PONE-D-21-13771R1Pregnant women’s intentions and factors associated with maternity waiting home utilization in rural districts of Hadiya Zone, Southern EthiopiaPLOS ONE

Dear Dr. Mosa,

Thank you for submitting your manuscript to PLOS ONE. After careful consideration, we feel that it has merit but does not fully meet PLOS ONE’s publication criteria as it currently stands. Therefore, we invite you to submit a revised version of the manuscript that addresses the points raised during the review process.

The manuscript has been evaluated by two reviewers, and their comments are available below.

The reviewers have raised a number of concerns that need attention.

Could you please revise the manuscript to carefully address the concerns raised?

We look forward to receiving your revised manuscript.

Kind regards,

Sebastian Shepherd

Associate Editor

PLOS ONE

Journal Requirements:

Reviewers' comments:

Reviewer's Responses to Questions

**Comments to the Author**

1. If the authors have adequately addressed your comments raised in a previous round of review and you feel that this manuscript is now acceptable for publication, you may indicate that here to bypass the “Comments to the Author” section, enter your conflict of interest statement in the “Confidential to Editor” section, and submit your "Accept" recommendation.

Reviewer #1: All comments have been addressed

Reviewer #3: (No Response)

2. Is the manuscript technically sound, and do the data support the conclusions?

Reviewer #1: Yes

Reviewer #3: Yes

3. Has the statistical analysis been performed appropriately and rigorously? 

Reviewer #1: Yes

Reviewer #3: No

4. Have the authors made all data underlying the findings in their manuscript fully available?

Reviewer #1: Yes

Reviewer #3: Yes

5. Is the manuscript presented in an intelligible fashion and written in standard English?

Reviewer #1: Yes

Reviewer #3: No

6. Review Comments to the Author

Reviewer #1: This article adds to the growing literature on maternity waiting homes internationally and specifically in Ethiopia. The authors have addressed all previous comments from reviewers. It is a much stronger article and ready for publication.

Reviewer #3: Major comments;

1. Researchers used systematic sampling technique to recruit respondents. But it is not clear how the sampling frame was established? Did you do household listing? And subsequent household was contacted in cases of no pregnant woman in the selected household. It means that every pregnant woman did not have an equal chance of selection and induces bias.

2. Most previous studies identified maternal education and distance to health facility predicts use of/ intention to use MWHs. But in this study these variables are not included in the multivariable analysis. Please force retain them in the multivariable analysis and discuss and interpret possible lack of association with the outcome variable

3. Number of ANC visit would possible be confounded or modified by gestational age. Please add gestation age in the analysis and possible interaction term. Otherwise, interpretation of # of ANC would be misleading and ambiguous

4. Multiple grammatical errors that need to be fixed.

Minor comments;

1. The Title seems a qualitative study design. I would drop “Facilitators and barriers” and replace associated or predictors factors

2. Line 57, replace majority by most

3. Line 58-61, update by MiniDHS 2019 findings

4. Line 69, remove, between that and MWHs

5. Line 72-73, contradictory evidence regarding wealth. Check that

6. Paragraph 3 is too long and difficult to catch. Reorganize/theme factors and narrate in a separate paragraph

7. Line 92-93, significance of the study is a bit exaggerated. Discount to … would help program managers and service providers to design appropriate interventions….and provide appropriate care for pregnant women….

8. Sub-title the settings, design, sample size and sampling techniques

9. Line 101, define kebele here; that means move the definition in line 105 here

10. Line 123, how fundal height is measured at home?

11. Line 124-127, move it to “Measurement” section

12. In the definition of terms like positive subjective norms, please give items included to measure such terms

13. Line 147, report exact value of alpha

14. Line 147, what are the local languages used? Hadiyigna and Amharic? Please specify

15. Line 149, change hadiya to Hadiya. Same comment throughout the manuscript

16. Line 150, specify what changes did researchers made after pre-testing

17. Line 179, what type of treatment? This is confusing as this is a household survey.

18. Line 183, replace majority by most

19. Line 182, as researchers replace respondents/household, I don’t think there is a need to report response rate.

20. Line 189-190, specify obstetric complication and ANC4+ are from current or past pregnancy

21. Table 1, categorize marital status as in union and not in union; number of households not clear if it is to mean number of household members correct as such; time taken to reach facility in minutes; specify means of transport (it is walking, by car, etc.)

22. Table 2, number of ANC visits ever received; change still birth to stillbirth

23. Discussion section, use odds of… rather than more likely…

24. Line 314-15, recall bias is a problem not only for the outcome variable but also for the predictor variables too. Please also add social desirability bias and mention techniques you applied to minimize such biases

25. Line 329, please acknowledge Hossana College of Health Science for funding the study

7. PLOS authors have the option to publish the peer review history of their article (what does this mean?). If published, this will include your full peer review and any attached files.

Reviewer #1: No

Reviewer #3: **Yes: **Gizachew Tadele Tiruneh

---

## [Author Response · Author response to Decision Letter 1]

8 Apr 2022

My coauthors and I thank you for your comments and suggestions concerning our manuscript “Pregnant women’s intentions and factors associated with maternity waiting home utilization in rural districts of Hadiya Zone, Southern Ethiopia”). We also genuinely appreciate the thoughtful comments from the reviewers, which have improved the paper.

We have studied the comments carefully and have revised our paper accordingly. This letter provides point-by-point responses to each comment and summarizes relevant changes in the manuscript. These changes are highlighted yellow in the revised manuscript. Primary changes to the paper include the following:

The reviewers have raised a number of concerns that need attention.

Could you please revise the manuscript to carefully address the concerns raised?

Response to Editor: Thank you for your time and consideration. We have addressed each reviewer comment and have revised the manuscript according to the reviewers’ comments.

Journal Requirements: Please review your reference list to ensure that it is complete and correct. If you have cited papers that have been retracted, please include the rationale for doing so in the manuscript text, or remove these references and replace them with relevant current references. Any changes to the reference list should be mentioned in the rebuttal letter that accompanies your revised manuscript. If you need to cite a retracted article, indicate the article’s retracted status in the References list and also include a citation and full reference for the retraction notice.

Response: We have reviewed and adjusted our references to ensure that they are in compliance with PLOS specifications. All changes are highlighted yellow in the revised manuscript. The editor may see the reference part. 

Review Comments to the Author

Reviewer #3: Major comments;

1. Researchers used systematic sampling technique to recruit respondents. But it is not clear how the sampling frame was established? Did you do household listing? And subsequent household was contacted in cases of no pregnant woman in the selected household. It means that every pregnant woman did not have an equal chance of selection and induces bias.

Response: Thank you for your comment. Now we have modified it based on the given comment. The reviewer may see the method part.

2. Most previous studies identified maternal education and distance to health facility predicts use of/ intention to use MWHs. But in this study these variables are not included in the multivariable analysis. Please force retain them in the multivariable analysis and discuss and interpret possible lack of association with the outcome variable

Response: Thank you for the comment. We have amended it according to the given comment.

3. Number of ANC visit would possible be confounded or modified by gestational age. Please add gestation age in the analysis and possible interaction term. Otherwise, interpretation of # of ANC would be misleading and ambiguous

4. Multiple grammatical errors that need to be fixed.

Response: Thank you again for bringing this to our attention. We have also obtained the help of an editor to correct errors in grammar and sentence structure throughout the manuscript. Every change in the manuscript has been highlighted in yellow.

Minor comments;

1. The Title seems a qualitative study design. I would drop “Facilitators and barriers” and replace associated or predictors factors

Response: The title is already modified by the previous comment as (Pregnant women’s intentions and factors associated with maternity waiting home utilization in rural districts of Hadiya Zone, Southern Ethiopia). The reviewer may see the title.

2. Line 57, replace majority by most

Response: Thank you for the comment. Now we have made the needed change.

3. Line 58-61, update by MiniDHS 2019 findings

Response: Thank you for your time and consideration. We have updated the sentences in the introduction part.

4. Line 69, remove, between that and MWHs

Response: Thank you for the comment. Now we have corrected it according to the given comment.

5. Line 72-73, contradictory evidence regarding wealth. Check that

Response: Thank you for your deep insight. We understand the raised concern. But, it is correct since we have checked it.

6. Paragraph 3 is too long and difficult to catch. Reorganize/theme factors and narrate in a separate paragraph

Response: Thank you for the comment. Correction is made according to the raised concern. The reviewer may see the introduction part.

7. Line 92-93, significance of the study is a bit exaggerated. Discount to … would help program managers and service providers to design appropriate interventions….and provide appropriate care for pregnant women….

Response: Thank you for bringing this to our attention. We have updated the sentences in the introduction part.

8. Sub-title the settings, design, sample size and sampling techniques

Response: Thank you for the comment. Now we have made the essential correction. The reviewer may see the method part

9. Line 101, define kebele here; that means move the definition in line 105 here

Response: Thank you for the comment. Now we have made the necessary amendment.

10. Line 123, how fundal height is measured at home?

Response: Thank you for the remark. Now we have made the necessary modification.

11. Line 124-127, move it to “Measurement” section

Response: We are indebted for the provided comment. We have made the necessary amendment according to the given comment. The reviewer may the measurement part.

12. In the definition of terms like positive subjective norms, please give items included to measure such terms

13. Line 147, report exact value of alpha

Response: Thank you for the comment. We accept the suggestion and we have mentioned the exact value of alpha. The reviewer may see the method part.

14. Line 147, what are the local languages used? Hadiyigna and Amharic? Please specify

Response: We are grateful you for the provided comment. Now we have made the required change the reviewer may see the method part.

15. Line 149, change hadiya to Hadiya. Same comment throughout the manuscript

Response: Thank you very much for your comment. Now the method part is corrected as per the given suggestion. The reviewer may see the method part.

16. Line 150, specify what changes did researchers made after pre-testing

Response: We are thankful for the delivered statement and we have made an essential modification in this regard. The reviewer may see the method part.

17. Line 179, what type of treatment? This is confusing as this is a household survey.

Response: We are grateful you for the provided comment. Now we have made the required change by removing the term household survey. The reviewer may see the method part.

18. Line 183, replace majority by most

Response: Thank you very much for the given comment. We have corrected it according to the given comment.

19. Line 182, as researchers replace respondents/household, I don’t think there is a need to report response rate.

Response: Thank you very much for the given comment. Now we have modified it.

20. Line 189-190, specify obstetric complication and ANC4+ are from current or past pregnancy

Response: Thank you for the comment. Now we included definition for obstetric complication in measurement part besides, the necessary correction is made with regard to ANC4+.

21. Table 1, categorize marital status as in union and not in union; number of households not clear if it is to mean number of household members correct as such; time taken to reach facility in minutes; specify means of transport (it is walking, by car, etc.)

Response: We are indebted for the provided statement and we have made the essential modification in this regard. The reviewer may see table 1.

22. Table 2, number of ANC visits ever received; change still birth to stillbirth

Response: We are appreciative for the provided comment and we have made the necessary amendment in this regard. 

23. Discussion section, use odds of… rather than more likely…

Response: Thank you for pointing this out. We have made the requested changes. The reviewer may see the discussion part.

24. Line 314-15, recall bias is a problem not only for the outcome variable but also for the predictor variables too. Please also add social desirability bias and mention techniques you applied to minimize such biases

Response: We are thankful for the provided comment and we have made the necessary modification on the limitation part. The reviewer may see this part.

25. Line 329, please acknowledge Hossana College of Health Science for funding the study

Response: We are grateful for the given comment and we have made the essential modification on the acknowledgement part. The reviewer may see this part.

---

## [Editor Report · Decision Letter 2]

4 May 2022

PONE-D-21-13771R2

Pregnant women’s intentions and factors associated with maternity waiting home utilization in rural districts of Hadiya Zone, Southern Ethiopia

PLOS ONE

Dear Dr. Mosa,

Thank you for submitting your manuscript to PLOS ONE. After careful consideration, we feel that it has merit but does not fully meet PLOS ONE’s publication criteria as it currently stands. Therefore, we invite you to submit a revised version of the manuscript that addresses the points raised during the review process.

Reviewers have continued to note that this manuscript could make an important contribution to the literature on maternity waiting homes. I had the pleasure of reviewing the manuscript myself during its initial evaluation. After assessing your most recent response to reviewers, I have identified multiple items raised by Reviewer 3 that have not been sufficiently addressed in your resubmission. These require a Major Revision to the manuscript. Please address the below listed items and resubmit the manuscript. I have retained Reviewer #3's original numbering below for ease of comparing with your most recent response to reviewers. I have included a few clarifying notes within parentheses.

Unaddressed Points Raised by Reviewer #3:

1. Researchers used systematic sampling technique to recruit respondents. But it is not clear how the sampling frame was established? Did you do household listing?

2. Most previous studies identified maternal education and distance to health facility predicts use of/ intention to use MWHs. But in this study these variables are not included in the multivariable analysis. Please force retain them in the multivariable analysis and discuss and interpret possible lack of association with the outcome variable.

3. Number of ANC visit would possible be confounded or modified by gestational age. Please add gestation age in the analysis and possible interaction term. Otherwise, interpretation of # of ANC would be misleading and ambiguous

9. Line 101, define kebele here; that means move the definition in line 105 here (i.e. define Kebele the first time it is mentioned in the text)

10. Line 123, how fundal height is measured at home? (i.e. gestational age)

12. In the definition of terms like positive subjective norms, please give items included to measure such terms (ensure items used to construct this indicator are explained within the methods)

17. Line 179, what type of treatment? This is confusing as this is a household survey. (Is this treatment at the health facility as the interviewers were midwives? This requires explanation within the text)

I concur with Reviewer #3 regarding the need to clarify gestational age within the analysis as gestational age does not appear to have been accounted for. If gestation age was an eligibility criteria it should be described in the methods as such. If it was not considered within the study, that should be explained in the limitations. I further concur with Reviewer #3 that "positive subjective norms" requires further explanation within the methods. While "5 items" is stated on Line 146, information on what those five items are is not provided.

In addition, as PLOS ONE does not offer copy-editing services at the final stage of publication, this manuscript still requires some English editing for sentence structure, grammar, and word choice. The manuscript also requires editing for minor formatting issues (e.g. double spaces within tables, incorrectly located header for Table 3, the spelling of "newborn" without a hyphen).

We look forward to receiving your revised manuscript.

Kind regards,

Jeanette L Kaiser, MPH

Guest Editor

PLOS ONE
---

## [Author Response · Author response to Decision Letter 2]

30 Jun 2022

Thank you very much for your comments, suggestions and critical appraisal.

 PONE-D-21-13771R2

Pregnant women’s intentions and factors associated with maternity waiting home utilization in rural districts of Hadiya Zone, Southern Ethiopia 

Thank you for submitting your manuscript to PLOS ONE. After careful consideration, we feel that it has merit but does not fully meet PLOS ONE’s publication criteria as it currently stands. Therefore, we invite you to submit a revised version of the manuscript that addresses the points raised during the review process.

Reviewers have continued to note that this manuscript could make an important contribution to the literature on maternity waiting homes. I had the pleasure of reviewing the manuscript myself during its initial evaluation. After assessing your most recent response to reviewers, I have identified multiple items raised by Reviewer 3 that have not been sufficiently addressed in your resubmission. These require a Major Revision to the manuscript. Please address the below listed items and resubmit the manuscript. I have retained Reviewer #3's original numbering below for ease of comparing with your most recent response to reviewers. I have included a few clarifying notes within parentheses.

Unaddressed Points Raised by Reviewer #3:=

1. Researchers used systematic sampling technique to recruit respondents. But it is not clear how the sampling frame was established? Did you do household listing?

Response. Thank you for the comment. Yes, we have performed household listing the sampling frame. Now we have made a considerable amendment. The reviewer may see the method part.

2. Most previous studies identified maternal education and distance to health facility predicts use of/ intention to use MWHs. But in this study these variables are not included in the multivariable analysis. Please force retain them in the multivariable analysis and discuss and interpret possible lack of association with the outcome variable.

Response. We are thankful for the given comment. Now we have interpreted possible lack of association for maternal education and distance to health facility in the discussion part. The reviewer may see the discussion portion.

 3. Number of ANC visit would possible be confounded or modified by gestational age. Please add gestation age in the analysis and possible interaction term. Otherwise, interpretation of # of ANC would be misleading and ambiguous

Response. We are thankful for the given comment. We have added gestation age. The reviewer may see the measurement part.

9. Line 101, define kebele here; that means move the definition in line 105 here (i.e. define Kebele the first time it is mentioned in the text)

Response. Thank you for the comment. Correction is made according to the provided comment. 

10. Line 123, how fundal height is measured at home? (i.e. gestational age)

Response. Thank you for the comment. We have shared the concern. Fundal height is measured since the data collectors are midwives.

12. In the definition of terms like positive subjective norms, please give items included to measure such terms (ensure items used to construct this indicator are explained within the methods).

Response. Thank you for the comment. At this instant, we have included the items used to measure subjective norms. You may see the measurement part.

17. Line 179, what type of treatment? This is confusing as this is a household survey. (Is this treatment at the health facility as the interviewers were midwives? This requires explanation within the text)

Response. We are grateful for the given comment and we have removed this word.

I concur with Reviewer #3 regarding the need to clarify gestational age within the analysis as gestational age does not appear to have been accounted for. If gestation age was an eligibility criteria it should be described in the methods as such. If it was not considered within the study, that should be explained in the limitations.

Response. We are thankful for the given comment. In this study, gestational age was considered. For determining gestational age we have used last menstrual period and fundal height as the data collectors are midwives.

I further concur with Reviewer #3 that "positive subjective norms" requires further explanation within the methods. While "5 items" is stated on Line 146, information on what those five items are is not provided. 

Response. We are thankful for the given comment. Now we have included the items used to measure subjective norms. You may see the measurement part. In addition, as PLOS ONE does not offer copy-editing services at the final stage of publication, this manuscript still requires some English editing for sentence structure, grammar, and word choice. The manuscript also requires editing for minor formatting issues (e.g. double spaces within tables, incorrectly located header for Table 3, the spelling of "newborn" without a hyphen).

Response. Thank you for the comment. Now we have used professional editing services like paper true and proofreading to correct spelling, punctuation and grammatical errors. Formatting issues were considered according to the given comment.

---

## [Decision Letter · Decision Letter 3]

1 Sep 2022

PONE-D-21-13771R3Pregnant women’s intentions and factors associated with maternity waiting home utilization in rural districts of Hadiya Zone, Southern EthiopiaPLOS ONE

Dear Dr. Mosa,

Thank you for submitting your manuscript to PLOS ONE. After careful consideration, we feel that it has merit but does not fully meet PLOS ONE’s publication criteria as it currently stands. Therefore, we invite you to submit a revised version of the manuscript that addresses the points raised during the review process.

We look forward to receiving your revised manuscript.

Kind regards,

Jeanette L Kaiser, MPH

Guest Editor

PLOS ONE

Additional Editor comments:

Thank you for your submitted revisions. Comments from the reviewers are included below. The reviewers advise additional copy editing of the manuscript for grammar, spelling, and spacing. As PLOS ONE does not provide copy editing services before publication, this is the responsibility of the authors. Please ensure all grammatical, spelling, and formatting errors are corrected with resubmission of your manuscript.

Reviewers' comments:

Reviewer's Responses to Questions

**Comments to the Author**

1. If the authors have adequately addressed your comments raised in a previous round of review and you feel that this manuscript is now acceptable for publication, you may indicate that here to bypass the “Comments to the Author” section, enter your conflict of interest statement in the “Confidential to Editor” section, and submit your "Accept" recommendation.

Reviewer #2: (No Response)

Reviewer #3: All comments have been addressed

2. Is the manuscript technically sound, and do the data support the conclusions?

Reviewer #2: Yes

Reviewer #3: Yes

3. Has the statistical analysis been performed appropriately and rigorously? 

Reviewer #2: Yes

Reviewer #3: Yes

4. Have the authors made all data underlying the findings in their manuscript fully available?

Reviewer #2: Yes

Reviewer #3: Yes

5. Is the manuscript presented in an intelligible fashion and written in standard English?

Reviewer #2: No

Reviewer #3: No

6. Review Comments to the Author

Reviewer #2: Grammatical, spelling and spacing errors throughout the manuscript.

In response to reviewer #3, the authors have added definitions of terms but they seem misplaced in the manuscript. I would recommend using a secondary section in the methods section where these definitions are listed and not just dropped into the methods section.

Please add information on the informed consent process of participants within the data collection description for better flow.

How is “wealth status” calculated/defined?

Line 297-299: “This finding implies that, as long as the issue of service quality is addressed, women in the wealthier quintiles can be encouraged to use MWHs.” I don’t believe the authors provide the data to support this statement.

Reviewer #3: Minor comments

1. This paper would benefit from an academic editor

2. Fix grammatical errors such as line 84, change the word moms to mothers or women; line 85, don't hyphenate newborns; line 92, remove double ","; line 116, change the word are by were and specify the period; line 120 change the word Health Centers to health centers; line 133-134, the sentence is not clear ( is that to mean proportional allocation was made to Kebeles?); line 149 drop "for institutional"; line 169 and 173 change the word was to were; line 219 change multifarious to multiparous; line 309 change the word above to more

3. Line 129, the source population is 1.65 million; why the authors applied finite population correction is not clear.

4. Line 134-137, the sampling interval is not specified; still it is not clear how researchers locate households once systematically selected. Did researchers use local guiders or HEWs to locate households? Please specify

5. Drop some of the operational definitions like SFH; nothing adds to this paper.

6. Line 225, indicate these percentages are from those who intend to stay at MWHs, not from the whole sample

7. Table 3, the values for the variable "distance to health facility" is not correct; please fix it; also add percentage values

7. PLOS authors have the option to publish the peer review history of their article (what does this mean?). If published, this will include your full peer review and any attached files.

Reviewer #2: No

Reviewer #3: No

---

## [Author Response · Author response to Decision Letter 3]

3 Oct 2022

First of all, we want to thank you very much for your detail questions, comments and suggestions.

PONE-D-21-13771R3

Pregnant women’s intentions and factors associated with maternity waiting home utilization in rural districts of Hadiya Zone, Southern Ethiopia

Additional Editor Comments:

Thank you for your submitted revisions. Comments from the reviewers are included below. The reviewers advise additional copy editing of the manuscript for grammar, spelling, and spacing. As PLOS ONE does not provide copy editing services before publication, this is the responsibility of the authors. Please ensure all grammatical, spelling, and formatting errors are corrected with resubmission of your manuscript.

Response to editor: We are very grateful for the comment you provided. Now, as much as possible, we have tried to correct language usage flaws including punctuations, wordings, spelling and grammar errors using English editing service.

Reviewers' comments:

Reviewer #2: Grammatical, spelling and spacing errors throughout the manuscript.

Response to reviewer: We are grateful for the suggestion. At this instant we have used English editing service to correct spelling, punctuation and grammatical errors. The reviewer may see the yellow highlighted part mainly on the introduction and discussion part.

In response to reviewer #3, the authors have added definitions of terms but they seem misplaced in the manuscript. I would recommend using a secondary section in the methods section where these definitions are listed and not just dropped into the methods section.

Response to reviewer: We are very thankful for the comment you delivered. In this instant we have written in a secondary section of method part. The reviewer may see the method part on page 7 and 8. 

Please add information on the informed consent process of participants within the data collection description for better flow. 

Response to reviewer: Thank you for the provided comment. By understanding your concern, we have modified the informed consent process to make it coherent. The reviewer may see the method part on page 9.

How is “wealth status” calculated/defined?

Response to reviewer: The household wealth variable was created using principal components analysis of items listed in below. Items were selected to minimize clustering and truncation which compromise reliability An asset-based wealth index created using information on asset ownership (radio, television, mobile phone, motorbike, car/truck), number of animals owned (cows, sheep, poultry), electricity supply to home, health insurance, drinking water source, type of toilet and type materials used for construction of floors in the home. It was divided into three groups: low (poor), medium, and high (rich) [26]. 

26. Vyas S, Kumaranayake L. Constructing socio-economic status indices: how to use principal components analysis. Health Policy Plan 2006; 21:459-68. The reviewer may see the method part on page 9.

Line 297-299: “This finding implies that, as long as the issue of service quality is addressed, women in the wealthier quintiles can be encouraged to use MWHs.” I don’t believe the authors provide the data to support this statement.

Response to reviewer: Thank you for the given comment. We have understood the raised concern. Now we have corrected agreeing to the provided comment. According to the findings of the current study, poor pregnant women have 2.52 times higher probabilities of intending to use MWHs than their counterparts. This is in line with a study carried out in Tanzania [11]. The reason could be related to the fact that for poorer women, the MWHs may be a means to access higher level obstetric care without incurring costs for private transport during labor and delivery. However, this result is in contrast with the studies conducted in Northwest Ethiopia, Jimma zone, and Timor Leste, respectively [8, 9, 23]. The reason could be the wealthiest women have the economic means for emergency transport to the hospital once labor starts. The reviewer may see the discussion part.

8. Endayehu M, Yitayal M, Debie A. Intentions to use maternity waiting homes and associated factors in Northwest Ethiopia. BMC Pregnancy Childbirth.2020;20(1):1-10. https://doi.org/10.1186/s12884-020-02982-0.

9. Kurji J, Gebretsadik LA, Wordofa MA, et al Factors associated with maternity waiting home use among women in Jimma Zone, Ethiopia: a multilevel cross-sectional analysis BMJ Open 2019;9:e028210. https://bmjopen.bmj.com/content/9/8/e028210

23. Wild K, Barclay L, Kelly P, Martins N. The tyranny of distance: maternity waiting homes and access to birthing facilities in rural Timor-Leste. Bull World Health Organ. 2012; 90:97- 103. https://doi.org/10.2471/BLT.11.088955. PMID: 22423160

Reviewer #3: Minor comments

1. This paper would benefit from an academic editor

Response to reviewer: We are very grateful for the comment you provided. Now, as much as possible, we have tried to correct language usage flaws including punctuations, wordings, spelling and grammar errors using English editing service. The reviewer may see the yellow highlighted part mainly on the introduction and discussion part.

2. Fix grammatical errors such as line 84, change the word moms to mothers or women; 

Response to reviewer: Now we have replaced the word mothers by moms. The reviewer may see the introduction part on page 5.

Line 85, don't hyphenate newborns; 

Response to reviewer: Now we have removed the hyphen. The reviewer may see the introduction part which is highlighted in yellow color.

Line 92, remove double ","; .

Response to reviewer: Now we have removed the hyphen. The reviewer may see the introduction part which is highlighted in yellow color.

Response to reviewer: Now we have removed the unnecessary commas. The reviewer may see the introduction part.

 Line 116, change the word are by were and specify the period;

 Response to reviewer: We have made correction according to the comment. The reviewer may see the method part on page 6.

Line 120 change the word Health Centers to health centers;

 Response to reviewer: We have made a correction by avoiding unnecessary capitalization. The reviewer may see page 6. 

Line 133-134, the sentence is not clear (is that to mean proportional allocation was made to Kebeles?);

 Response to reviewer: Proportional allocation was made to the number of households. The k-value of four was obtained by dividing the source population (1540) by the sample size (385). The reviewer may see page 6 on the method part for more clarification.

 Line 149 drop "for institutional"; 

Response to reviewer: We have accepted the comment and dropped in to new line. The reviewer may see page 6. 

Line 169 and 173 change the word was to were; 

Response to reviewer: We have made a correction. The reviewer may see page 8.

Line 219 change multifarious to multiparous; 

Response to reviewer: We have replace the word multiparous by multifarious. The reviewer may see page 10.

Line 309 change the word above to more.

Response to reviewer: we have replaced the word above by more. The reviewer may see page 6 

3. Line 129, the source population is 1.65 million; why the authors applied finite population correction is not clear. 

Response to reviewer: As stated by the reviewer, 1.65 million is not a source population. It is an overall population found in Hadiya zone. Instead, a total of 1540 pregnant women were reported as a source population from selected kebeles. That’s why we have used a finite population correction formula. The reviewer may see page 6 on method part.

4. Line 134-137, the sampling interval is not specified; still it is not clear how researchers locate households once systematically selected. Did researchers use local guiders or HEWs to locate households? Please specify

Response to reviewer: Proportional allocation was made to the number of households. The sampling interval (k-value) of four was obtained by dividing the source population (1540) by the sample size (385). We have used HEWs to locate households. The reviewer may see the method part on page 6.

5. Drop some of the operational definitions like SFH; nothing adds to this paper.

Response to reviewer: We have added the SFH as it was asked by other reviewer during the previous revisions that is why we have incorporated it.

6. Line 225, indicate these percentages are from those who intend to stay at MWHs, not from the whole sample 

Response to reviewer: Thank you for your deep insight. We understand the raised concern. Now we have calculated the percentage of favorable attitude from women who had an intention to use MWH not from overall intention to use MWH). We have made the necessary modification according to the given comment. The reviewer may see page 10 on the result part and table 3.

7. Table 3, the values for the variable "distance to health facility" is not correct; please fix it; also add percentage values

Response to reviewer: We have checked the value of the variable mention which is correct. But, we forgot to incorporate its percentage. Now we included its percentage. The reviewer may see table 3.

---

## [Decision Letter · Decision Letter 4]

14 Dec 2022

PONE-D-21-13771R4Pregnant women’s intentions and factors associated with maternity waiting home utilization in rural districts of Hadiya Zone, Southern EthiopiaPLOS ONE

Dear Dr. Mosa,

Thank you for submitting your manuscript to PLOS ONE. After careful consideration, we feel that it has merit but does not fully meet PLOS ONE’s publication criteria as it currently stands. Therefore, we invite you to submit a revised version of the manuscript that addresses the points raised during the review process.

ACADEMIC EDITOR: The reviewers have noted that this manuscript has made great progress but still requires English language proofing and editing. Reviewer #3 has provided specific instances where the writing can be clarified and improved. Please make these minor changes and resubmit.

We look forward to receiving your revised manuscript.

Kind regards,

Jeanette L Kaiser, MPH

Guest Editor

PLOS ONE

Journal Requirements:

Reviewers' comments:

Reviewer's Responses to Questions

**Comments to the Author**

1. If the authors have adequately addressed your comments raised in a previous round of review and you feel that this manuscript is now acceptable for publication, you may indicate that here to bypass the “Comments to the Author” section, enter your conflict of interest statement in the “Confidential to Editor” section, and submit your "Accept" recommendation.

Reviewer #3: (No Response)

Reviewer #4: (No Response)

2. Is the manuscript technically sound, and do the data support the conclusions?

Reviewer #3: Yes

Reviewer #4: Yes

3. Has the statistical analysis been performed appropriately and rigorously? 

Reviewer #3: Yes

Reviewer #4: Yes

4. Have the authors made all data underlying the findings in their manuscript fully available?

Reviewer #3: Yes

Reviewer #4: Yes

5. Is the manuscript presented in an intelligible fashion and written in standard English?

Reviewer #3: No

Reviewer #4: Yes

6. Review Comments to the Author

Reviewer #3: Dear authors,

• Title: change to "Pregnant women’s intentions to use maternity waiting homes and its associated factors in rural districts of Hadiya Zone, Southern Ethiopia"

• Still, this needs to be edited by an academic editor. Please also fully address the comments. For instance, the following grammatical errors are not addressed: line 84, change the word “moms” to “mothers or women”; line 85, don't hyphenate newborns; line 116, change the word “are” to “were” and specify the period; line 149 drop "for institutional"; line 170 change the word “was” to “were”;

• Line 26, change the word "raise" to "expand"; line 28, drop the phrase "across the world"' line 69, drop the word "dangerously"; line 75-78, change by "Furthermore, only about half women were delivered by a skilled provider and 34% of women received a postnatal care check-up within the first two days after birth [3]."; line 76, change "Antenatal" to "antenatal"; line 166, drop the word "LMP";

• Lines 92-93, drop "giving birth in a health facility, receiving skilled care from a trained provider [17, 19]" No temporal relationship between use of MWHs and skilled delivery, unless we asked for previous skilled assistance.

• Table 3, the percentage values for education, distance to health facility, and attitude could not add up to 100%; please fix it by adding column percentages similar to other variables

• Line 132-33, The sentence “Proportional allocation was made to the number of households.” is not clear and how many kebeles were included in the study is not clear. Please replace this with the following sentences in your original manuscript “A lottery method was used to pick ten rural kebeles. According to the approximate number of pregnant women in each kebele, the sample size was distributed proportionally.”

• Check again or consult a statistician about the finite population correction. I think it is applied when the target population (i.e., the entire pregnant women for which the survey data are to be used to make inferences) not the source population (i.e., pregnant women residing in the PSU) is finite

Regards,

Reviewer #4: The document has improved greatly. The authors can improve the document by correcting typo

and grammatical errors and proofing

7. PLOS authors have the option to publish the peer review history of their article (what does this mean?). If published, this will include your full peer review and any attached files.

Reviewer #3: **Yes: **Gizachew Tadele Tiruneh

Reviewer #4: **Yes: **Dr Sialubanje Cephas,

School of Public Health

Levy Mwanawasa Medical University

Lusaka, Zambia

csialubanje@yahoo.com

---

## [Author Response · Author response to Decision Letter 4]

1 Jan 2023

Pregnant women’s intentions to use maternity waiting homes and its associated factors in rural districts of Hadiya Zone, Southern Ethiopia

First of all, we want to thank you both the editor and reviewers for your detail questions, comments and suggestions. We strongly believe without your significant and valuable expertise the manuscript will not be in this stand.

PONE-D-21-13771R4

Additional Editor Comments:

Response to editor: Fist of all, we are grateful for your valuable comments and suggestions. We have checked and reviewed all the references list and we haven’t found problem with it. 

Reviewers' comments:

This paper would benefit from an academic editor

Response to reviewer 3. Always we wonder on the critical insight of the reviewer. Thank very much for all your effort and commitment. Now, as much as possible, we have tried to correct language usage flaws including punctuations, wordings, spelling and grammar errors using English editing service.

Title: change to "Pregnant women’s intentions to use maternity waiting homes and its associated factors in rural districts of Hadiya Zone, Southern Ethiopia"

Response to reviewer 3. We have agreed up the suggested modification on the title. The reviewer may see the title highlighted in yellow color.

• Still, this needs to be edited by an academic editor. Please also fully address the comments. For instance, the following grammatical errors are not addressed:

Response to reviewer 3. We have tried to improve language usage flaws including punctuations, wordings, spelling and grammar errors using English editing service.

 Line 84, change the word “moms” to “mothers or women”; 

Response to reviewer 3. We have replaced the word moms by mothers. The reviewer may see line- in the introduction part which is highlighted in yellow color.

Line 85, don't hyphenate newborns; 

Response to reviewer 3. By accepting the raised concern we have removed the hyphen. The reviewer may see the introduction part which is highlighted in yellow color.

 Line 116, change the word “are” to “were” and specify the period; 

Response to reviewer 3. The given remark is acknowledged and amendment has been made convening to the given recommendation. The reviewer may see the method part which is highlighted in yellow color.

Line 149 drop "for institutional";

Response to reviewer 3. The advised comment has been accepted and modified accordingly. The reviewer may see the method part highlighted in yellow color.

 Line 170 change the word “was” to “were”;

Response to reviewer 3. The recommended comment has been accepted and modified accordingly by replacing the word was by were. The reviewer may see the method part.

• Line 26, change the word "raise" to "expand"; 

Response to reviewer 3. Now we have replaced the word moms by mothers. The reviewer may see the abstract part highlighted in yellow color.

Line 28, drop the phrase "across the world"'

The proposed comment has been accepted and modified accordingly. The reviewer may see the abstract part highlighted in yellow color.

Line 69, drop the word "dangerously"; 

Response to reviewer 3. The suggested comment has been accepted and now totally we have removed it because its presence is not as such important. The reviewer may see the introduction part on page which is highlighted in yellow color.

 Line 75-78, change by "Furthermore, only about half women were delivered by a skilled provider and 34% of women received a postnatal care check-up within the first two days after birth [3].” 

Response to reviewer 3. The suggested comment has been accepted and modified accordingly. The reviewer may see the introduction part which is highlighted in yellow color.

Line 76, change "Antenatal" to "antenatal"; 

Response to reviewer 3. The recommended comment has been accepted and modified accordingly by replacing the word was by were. The reviewer may see the method part.

Line 166, drop the word "LMP";

Response to reviewer 3. We have accepted the comment and dropped in to new line. The reviewer may see the method part which is highlighted in yellow color. 

Lines 92-93, drop "giving birth in a health facility, receiving skilled care from a trained provider [17, 19]" No temporal relationship between use of MWHs and skilled delivery, unless we asked for previous skilled assistance.

Response to reviewer 3. We have accepted the comment and dropped in to new line. The reviewer may see the introduction part which is highlighted in yellow color

 Table 3, the percentage values for education, distance to health facility, and attitude could not add up to 100%; please fix it by adding column percentages similar to other variables

Response to reviewer 3. We have agreed up on the raised concern and made a necessary modification regarding percentage of the mentioned variables. The reviewer may see table 3 which is highlighted in yellow color.

Line 132-33, the sentence “Proportional allocation was made to the number of households.” is not clear and how many kebeles were included in the study is not clear. Please replace this with the following sentences in your original manuscript “A lottery method was used to pick ten rural kebeles. According to the approximate number of pregnant women in each kebele, the sample size was distributed proportionally.”

Response to reviewer 3. 

• Check again or consult a statistician about the finite population correction. I think it is applied when the target population (i.e., the entire pregnant women for which the survey data are to be used to make inferences) not the source population (i.e., pregnant women residing in the PSU) is finite

Response to reviewer 3. We have consulted a statistician regarding population correction formula. But, we have not received a different idea with the statistician in this regard.

---

## [Editor Report · Decision Letter 5]

30 Jan 2023

Pregnant women’s intentions to use maternity waiting homes and its associated factors in rural districts of Hadiya Zone, Southern Ethiopia

PONE-D-21-13771R5

Dear Dr. Mosa,

We’re pleased to inform you that your manuscript has been judged scientifically suitable for publication and will be formally accepted for publication once it meets all outstanding technical requirements.

Kind regards,

Jeanette L Kaiser, MPH

Guest Editor

PLOS ONE

Additional Editor Comments (optional):

Congratulations on the acceptance of you manuscript.

Reviewers' comments:

n/a

---

## [Editor Report · Acceptance letter]

25 May 2023

PONE-D-21-13771R5 

Pregnant women’s intentions to use maternity waiting homes and its associated factors in rural districts of Hadiya Zone, Southern Ethiopia 

Dear Dr. Mosa:

I'm pleased to inform you that your manuscript has been deemed suitable for publication in PLOS ONE. Congratulations! Your manuscript is now with our production department. 

Kind regards, 

on behalf of

Ms. Jeanette L Kaiser 

Guest Editor

PLOS ONE